# Preclinical Evaluation of Invariant Natural Killer T Cells Modified with CD38 or BCMA Chimeric Antigen Receptors for Multiple Myeloma

**DOI:** 10.3390/ijms22031096

**Published:** 2021-01-22

**Authors:** Renée Poels, Esther Drent, Roeland Lameris, Afroditi Katsarou, Maria Themeli, Hans J. van der Vliet, Tanja D. de Gruijl, Niels W. C. J. van de Donk, Tuna Mutis

**Affiliations:** 1Cancer Center Amsterdam, Department of Haematology, Amsterdam UMC, VU Amsterdam, 1081 HV Amsterdam, The Netherlands; r.poels@amsterdamumc.nl (R.P.); e.drent@gadeta.nl (E.D.); a.katsarou@amsterdamumc.nl (A.K.); m.themeli@amsterdamumc.nl (M.T.); n.vandedonk@amsterdamumc.nl (N.W.C.J.v.d.D.); 2Cancer Center Amsterdam, Department of Medical Oncology, Amsterdam UMC, VU Amsterdam, 1081 HV Amsterdam, The Netherlands; r.lameris@amsterdamumc.nl (R.L.); jj.vandervliet@amsterdamumc.nl (H.J.v.d.V.); td.degruijl@amsterdamumc.nl (T.D.d.G.); 3Lava Therapeutics, 3584 CM Utrecht, The Netherlands

**Keywords:** CAR iNKT, multiple myeloma, CD38, BCMA, adoptive immunotherapy, chimeric antigen receptor

## Abstract

Due to the CD1d restricted recognition of altered glycolipids, Vα24-invariant natural killer T (iNKT) cells are excellent tools for cancer immunotherapy with a significantly reduced risk for graft-versus-host disease when applied as off-the shelf-therapeutics across Human Leukocyte Antigen (HLA) barriers. To maximally harness their therapeutic potential for multiple myeloma (MM) treatment, we here armed iNKT cells with chimeric antigen receptors (CAR) directed against the MM-associated antigen CD38 and the plasma cell specific B cell maturation antigen (BCMA). We demonstrate that both CD38- and BCMA-CAR iNKT cells effectively eliminated MM cells in a CAR-dependent manner, without losing their T cell receptor (TCR)-mediated cytotoxic activity. Importantly, iNKT cells expressing either BCMA-CARs or affinity-optimized CD38-CARs spared normal hematopoietic cells and displayed a Th1-like cytokine profile, indicating their therapeutic utility. While the costimulatory domain of CD38-CARs had no influence on the cytotoxic functions of iNKT cells, CARs containing the 4-1BB domain showed a better expansion capacity. Interestingly, when stimulated only via CD1d^+^ dendritic cells (DCs) loaded with α-galactosylceramide (α-GalCer), both CD38- and BCMA-CAR iNKT cells expanded well, without losing their CAR- or TCR-dependent cytotoxic activities. This suggests the possibility of developing an off-the-shelf therapy with CAR iNKT cells, which might even be boostable in vivo by administration α-GalCer pulsed DCs.

## 1. Introduction

Over the past years, T cells engineered to express chimeric antigen receptors (CAR) have shown great successes in the battle against hematopoietic malignancies [1]. Nonetheless, there are still important hurdles to be taken in order to make CAR therapy more effective, safe, affordable, and universally accessible [1]. In particular, the necessity to use the patients’ autologous polyclonal T cells for the production of CAR T cells seem to generate a considerable variation in the efficacy of the therapy, since the quality of patient T cells can be greatly influenced by previous treatment schemes and/or the tumor type of the patients [1,2,3]. Hence, a patient-independent, off-the-shelf available CAR therapy is the ultimate goal of many investigators [4,5,6,7,8]. Application of CAR T cells across Human Leukocyte Antigen (HLA) barriers, however, carries the risk of severe graft-versus-host disease (GvHD) due to the expression of potentially alloreactive endogenous T cell receptors (TCRs) [9,10,11]. Therefore, several approaches have been proposed to silence the endogenous TCR expression of CAR T cells with novel gene editing technologies [12,13,14]. Alternatively, killer immune cell subsets are currently being explored as “universal” CAR carriers. An appealing candidate population consists of natural killer (NK) cells [15,16,17,18,19,20], as they are highly cytotoxic but are not associated with GvHD [21]. Furthermore, NK cells display a relatively short life-span which may allow better management of any toxicities of CAR-directed cellular therapies. On the other hand, building a long-term memory may not be possible with NK cells. An alternative is to introduce CARs into another subset of innate-like, but antigen-specific immune cells that can sustain a long-term memory, such as the invariant natural killer T (iNKT) cells [22]. iNKT cells recognize glycolipid antigens presented by highly conserved CD1d via their defined and invariant T cell receptor (TCR) (TRAV10-TRAJ18 paired with TRBV25-1 in humans) [23,24,25,26]. Due to this specific antigen-recognition pattern, iNKT cells are not associated with GvHD or can even prevent it [27,28]. Furthermore, iNKT cells possess therapeutic potential, their infiltration in tumor sites has been associated with better disease outcome [29], and reduced iNKT cell numbers and function have been associated with disease progression in multiple myeloma (MM) [30,31], prostate cancer [32], and several solid tumors [33].

Although iNKT cells represent a small fraction in the peripheral blood (0.003–0.7% of CD45^+^ cells) [34], they display strong TCR-dependent cytolytic function [35]. Since their CD1d-restricted TCR is responsive to α-galactosylceramide (α-GalCer), iNKT cells also demonstrate rapid ex vivo expansion when stimulated by CD1d positive antigen presenting cells loaded with α-GalCer [19,22,36]. Furthermore, in clinical studies, stimulation of iNKT cells with α-GalCer loaded antigen presenting cells enhanced their activity and decreased patients’ tumor load [37,38,39]. This TCR-specific expansion property makes iNKT cells appealing tools as CAR-carriers with low toxicity profiles. Thus, prompted by the therapeutic potential of iNKT cells, and especially by the possibility to expand and boost them through triggering their endogenous TCR, we here tested the feasibility of arming iNKT cells with CARs directed against two important MM-associated antigens, CD38- and the B cell maturation antigen (BCMA). For BCMA, we selected a high-affinity CAR which is already being tested in clinical trials [40]. For CD38, we have used a unique affinity-optimized (attenuated) CD38B1-CAR, which we developed in our laboratory. Due to its attenuated affinity, this CD38B1-CAR exclusively targets MM cells expressing high levels of CD38, while largely ignoring the healthy hematopoietic cells expressing normal levels of CD38 [41,42]. In earlier studies, we have found that the costimulatory domain of CD38-CARs can influence the effector and proliferative functions of T cells [43]. Therefore, we here tested three CD38B1-CAR formats providing different costimulatory signals. The first two CARs contained the traditional CD28 and 4-1BB costimulatory domains, respectively. The third CD38B1-CAR contained, next to the CD28 costimulatory domain, a separately expressed a 4-1BBL molecule. The separate expression of this molecule enables the additional triggering of the 4-1BB costimulatory pathway through autologous/paracrine 41BBL/41BB interactions [44,45].

Towards developing an off-the-shelf approach, we studied the CAR transduction efficacies, expansion capacities, the TCR- and CAR-dependent cytotoxic activity, and cytokine profiles against MM cells. To evaluate off-tumor effects, we tested their reactivity towards normal hematopoietic cells. In addition, we analyzed CAR iNKT cells for the possibility to expand through the endogenous TCR via stimulation with α-GalCer pulsed dendritic cells (DCs).

## 2. Results

### 2.1. Isolation and Transduction of Invariant NKT Cells

iNKT cells were isolated from healthy donor peripheral blood mononuclear cells (PBMCs) by Vα24^+^ selection using magnetic cell sorting. Directly after cell sorting the iNKT cell purity was ~30–70% with an iNKT cell content of 50.000–200.000 cells based on the double positive expression of Vα24 paired with Vβ11, as shown in Figure 1A. After one week of stimulation with α-galactosylceramide (α-GalCer) pulsed PBMCs, the cells expanded with a median of 11.7-fold (range 3–18), the purity increased up to 90%, and the cells were transduced with the affinity-optimized CD38B1-CARs containing three different co-stimulatory domains or with the BCMA-CAR, as schematically depicted in Figure 1B. The low-affinity nerve growth factor (LNGFR), dsRed, or 4–1BBL, which were co-expressed in the three different CD38-CAR constructs, were used to determine transduction efficiency, and used as a surrogate for CAR expression, as previously validated [43]. The BCMA-CAR expression was determined via direct CAR expression as measured by a goat-anti-mouse F(ab’)2 binding to the murine scFv sequence. The mean transduction efficacy of CAR-iNKT cells was ~50% (range: 20–90%, *n* = 8), as shown in Figure 1C, similar to transduction efficacies of conventional T cells in our earlier studies [42,43].

### 2.2. iNKT Cells Equipped with a CAR Show CAR-Specific as Well as TCR-Dependent Cytotoxicity

CAR-transduced iNKT cells were tested for their cytotoxic activity through the CAR-specific targeting of CD38 or BCMA expressed on multiple myeloma (MM) cell line UM9, as shown in Figure 2A. As expected, the UM9 cells were completely eradicated by the iNKT cells expressing the high affinity BCMA-CAR. Since the expression of CD38 on UM9 cells is intermediate, as shown in Figure 2A, left panel, a lysis up to 60% was observed for the affinity tuned CD38-CAR iNKTs, with no noteworthy distinction between CARs containing different costimulatory domains. Mock-transduced iNKT cells did not lyse UM9 cells.

To determine their cytotoxic activity via the CD1d-restriced invariant TCR, CD38-CAR, BCMA-CAR, and mock-transduced iNKT cells were tested against the CD1d intermediate positive MM1.s cells and against its CD1d-transduced variant with high levels of CD1d expression, as shown in Figure 2B. Since MM1.s cells express high levels of CD38 and BCMA, they were completely eliminated by both CD38- and BCMA-CAR iNKT cells even at low effector to target (E/T) ratios, whereas the lysis by mock-transduced iNKT cells was very low. Suggesting the intact signaling from the invariant TCR against MM cells, the mock-transduced cells killed the MM1.s cells up to 50% at high E/T ratios, in agreement with the intermediate CD1d expression detected on MM1.s, as shown in Figure 2C, left panel. Importantly, the CD1d-transduced MM1.s cells were completely eradicated, not only by CAR-transduced, but also mock-transduced iNKT cells even at low E/T ratios, suggesting the full functional activity of the endogenous CD1d restricted invariant TCR, as shown in Figure 2C, right panel.

### 2.3. Maximal On-Tumor and Minimal Off-Tumor Effects of CD38-CAR and BCMA-CAR Transduced iNKT Cells

To study the effect of CAR iNKT cells on primary MM cells, we conducted flow-based cytotoxicity assays on eight randomly selected bone marrow mononuclear cells (BM-MNC) from MM patients. These samples contained 10–40% malignant plasma cells (MM-PC) identified as cells expressing CD38^high^CD138^+^, as shown in Figure 3A,B. Since the BM-MNCs contain both malignant MM cells and non-malignant hematopoietic cells, this flow-based assay system allows us to simultaneously determine the on-tumor and off-tumor cytotoxic activity of CAR-transduced cells [41,42]. As illustrated in Figure 3C, all primary MM cells were effectively lysed by CD38-CAR as well as BCMA-CAR-transduced iNKT cells, with no influence of the costimulatory domain. Since BCMA is expressed only on PCs, BCMA-CAR iNKTs did not lyse CD138 negative non-malignant hematopoietic cells. Importantly, and consistent with all our previous studies [41,42,43], the cytotoxic activities of affinity optimized CD38-CAR iNKTs were also exclusively directed against CD38^high^CD138^+^ MM cells, while CD38^int/high^ CD138^-^ non-malignant cells were largely ignored, as shown in Figure 3A. Of note, in seven out of the eight samples there was little or no CD1d^+^ expression MM cells. Consequently, there was no mock-reactivity against these MM cells, as shown in Figure 3C, panel Mock. In the only sample that contained CD1d^+^ MM plasma cells there was significantly elevated levels of lysis by mock-transduced cells, again indicating the functional activity of the endogenous TCR, as shown in Appendix A.

### 2.4. The Cytokine Profiles CD38-CAR and BCMA-CAR-Transduced iNKT Cells

To further study the relevant functions of CAR iNKT cells, we analyzed the CAR-dependent cytokine production profile in response to one of the primary MM samples, as shown in Figure 4. A typical Th1 profile with very high levels of the proinflammatory cytokines IFN-γ and TNF but low or no production of Th2 cytokines IL-4 and IL-10 was found only for CD38-CARs carrying the 4-1BB costimulatory domain. Although other CAR iNKT cells also produced somewhat higher levels of IFN-γ as compared to IL-4, the IFN-γ to IL-4 ratios were in general lower than 10, suggesting a Th1/Th0 cytokine profile. Interestingly though, IL-2 was not detected above the levels observed from mock-transduced iNKT cells. A typical Th17 profile was not found in CAR iNKT cells. Furthermore, as expected, none of the CAR iNKT cells or mock-transduced iNKT cells produced significant levels of IL-6, one of the important cytokines associated with cytokine release syndrome.

### 2.5. CAR-Dependent Expansion of CD38-CAR iNKT Cells with Different Costimulatory Domains

We next evaluated the ex vivo expandability of the CAR iNKT cells. Since the costimulatory domain could influence the proliferative capacity, we focused on CD38-CAR iNKT cells and expanded them in a CAR-dependent manner by weekly stimulation with irradiated UM9 cells, as shown in Figure 5. The CD38-BBz-CAR iNKT cells vigorously proliferated upon stimulation with UM9, until the end of the experiment (Week 6), with total CAR^+^ iNKT cell yields around 10^7^ cells. The growth of CD38-CARs carrying the CD28 costimulatory domain stagnated after Week 4, as shown in Figure 5A. Mock-transduced cells proliferated to a lower extent as compared to CD38-BBz-CAR iNKT cells, revealing the contribution of CD38-specific stimulation in the exponential growth of CD38-CAR iNKT cells. Moreover, supporting this conclusion, at the end of the cultures, the percentage of CAR^+^iNKT cells in cultures transduced with CD38-BBz-CAR markedly increased from 30% to 70% and maintained their CAR-dependent cytotoxic activities, as shown in Figure 5C. iNKT cells transduced with the CD38-CAR carrying the 4-1BB costimulatory domain expressed little or low levels of PD-1 during the whole culture period. In contrast, CD38-CAR iNKT cells containing the CD28 costimulatory domain expressed high levels of PD-1, as shown in Figure 5D, left panel. In addition, the CD57 molecule, which is associated with exhaustion/senescence [46,47,48,49] was virtually absent in all CAR-transduced iNKT cells in contrast to mock-transduced cells, as shown in Figure 5D, middle panel. Reversely, the CD39 molecule, which is thought to regulate activation of iNKT cells [50,51], was positive on all CAR-transduced, but not on mock-transduced iNKT cells, as shown in Figure 5E, right panel. Taken together, these results indicated that 4-1BB provided the most suitable costimulatory signals for CAR-iNKT cells.

### 2.6. TCR-Mediated Proliferation of CD38-CAR and BCMA-CAR iNKT Cells

In further analyses, we evaluated whether the CAR iNKT cells could be stimulated and expanded by mature monocyte dendritic cells (moDC) loaded with α-GalCer. Since moDCs also express CD38, CD38-CAR iNKT cells would receive both CAR- and TCR-mediated signals, while BCMA-CARs would receive only signals from the invariant TCR. Therefore, we first studied the stimulation of BCMA-CARs in a series of experiments with four different donors, as shown in Figure 6A. BCMA-CAR iNKT cells vigorously proliferated upon stimulation with α-GalCer loaded DCs. Although their expansion rates were not as good as that of the mock-transduced iNKT cells, we achieved 1000-fold expansion of CAR^+^ iNKT cells in five weeks of time, as shown in Figure 6A, middle panel. As expected, the CAR iNKT percentages remained unchanged and BCMA CAR iNKT cells maintained their CAR-dependent cytotoxic activities at the end of the expansion period, as shown in Figure 6B.

Unlike the UM9 stimulated CD38-CARs, the BCMA-CARs stimulated with α-GalCer loaded DCs were largely positive for the PD-1 molecule during the whole culture period (data not shown). Therefore, in an independent setting we compared the growth of the iNKT cells transduced with BCMA-CARs and the most optimal CD38-BBz-CAR upon stimulation with α-GalCer loaded DCs, as shown in Figure 6C, and observed that the expansion rates of BCMA-CAR and CD38-CAR cells were somewhat lower (300–500-fold) but similar. Nonetheless, while the percentages of BCMA-CAR^+^ iNKT cells remained the same over time, the percentage of CD38-CAR^+^ iNKT cells diminished, as shown in Figure 6C, right panel. Furthermore, unlike CAR-dependent stimulations, CD38-CAR iNKT cells also expressed PD-1 after stimulation with α-GalCer loaded DCs. The checkpoint molecules Lag3 and Tim3 were also positive on both BCMA- and CD38-CAR iNKT cells.

Finally, we questioned whether CAR iNKT cells that have already been stimulated for several rounds with tumor cells in a CAR-dependent fashion could be further stimulated with α-GalCer loaded DCs. As illustrated in Figure 7, it was still possible to expand fully cytotoxic CD38-CAR iNKT cells by stimulation with α-GalCer loaded DCs, even if they had been stimulated for 6 weeks only with tumor cells.

## 3. Discussion

Over the past decade, several studies have convincingly demonstrated the anti-tumor potential of CD1d-restricted iNKT cells against different solid and hematological tumors [30,37,38,39]. It has been shown that iNKT cells not only produce cytokines but also effectively kill CD1d positive tumor cell lines [19,36]. Although iNKT cells are very low in frequency in human peripheral blood, the discovery that they can be exponentially expanded ex vivo or in vivo through stimulation with α-GalCer loaded dendritic cells (DCs) prompted the design of several strategies to utilize iNKT cells in cancer treatment. Nonetheless in MM, CD1d expression, which is high in premalignant and early disease, may be reduced and in some cases lost in the advanced stages [52], posing a potential drawback for iNKT-based immunotherapies. In this study, we have successfully tackled this drawback by endowing iNKT cells with a high affinity BCMA-CAR and with an affinity-optimized CD38-CAR, which selectively recognizes the high levels of CD38 antigen expressed on MM cells due to its attenuated CD38 affinity [43]. Both BCMA- and CD38-CAR-transduced iNKT cells effectively redirected the killer functions of iNKT cells toward MM cells. CD1d-transduced MM cells, which expressed high levels of CD1d were additionally killed, most probably through the recognition of endogenously expressed invariant TCR. Importantly, neither BCMA-CAR nor CD38-CAR-transduced iNKT cells displayed any significant off-tumor toxicity toward normal hematopoietic cells. We thus demonstrate that iNKT cells are excellent carriers for these therapeutically relevant CARs for adoptive therapy of MM.

In our study, we have not only focused on the effector function of CAR iNKT cells but have also given specific attention to define optimal conditions for their proper expansion. This was important since generation of sufficient numbers of CAR iNKT cells with intact CAR- and TCR-mediated functional activity would be required for adoptive therapy, especially if the ultimate aim is to develop an off-the-shelf therapy. In this respect, the correct choice of the costimulatory domain was especially important because we and others have recently found that costimulatory domains of CARs not only influence their cytotoxic activity but also their ex vivo and in vivo proliferation capacities [43]. Specifically, while a CD28 costimulatory domain has been shown to enhance the cytotoxic activity of CAR T cells, the 4-1BB signaling specifically enhanced the proliferative capacity and T cell persistence [43,45]. Nonetheless, when we studied the impact of the costimulatory domains using three variants of CD38-CARs, we did not observe an overt beneficial effect of the CD28 costimulatory domain on the cytotoxic activity of iNKT cells above CARs that expressed 4-1BB. In contrast, iNKT cells, equipped with a CAR carrying the 4-1BB domain showed a better CAR-driven expansion as compared to those carrying a CD28 domain, without any compromise in CAR-mediated cytotoxic activity, also after long-term (6–7 week) cultures. Importantly, in these long-term cultures, in contrast to CARs carrying the CD28 domain, CARs carrying the 4-1BB costimulatory domain did not induce the expression of exhaustion maker PD-1. The CD57 molecule, associated with maturation and immune senescence [46,47,48,49] was also not upregulated. CAR iNKTs that provided both CD28 and 4-1BB signals did not further improve their cytotoxic activity but were associated with diminished expansion capacities. These data are not entirely in agreement with earlier iNKT CAR studies, which suggests that for optimal stimulation of iNKT cells, CARs should include both 4-1BB or a CD28 costimulatory domain [19,53]. Furthermore, in contrast to these studies, the 4-1BB signaling skewed the CD38-CAR iNKT cells towards a Th1-like phenotype. Nonetheless, it needs to be noted that that BCMA-CARs, which also contained the 4-1BB costimulatory domain, skewed iNKT cells towards a Th1/Th0 phenotype. Thus, a clear association between the costimulatory domain and the Th1 phenotype cannot be derived from our results. It is noteworthy, however, that both CD38- and BCMA-CARs containing the 4-1BB domain did not skew iNKT cells towards IL-17-like cells and did not propagate the production of IL-6, one of the important cytokines associated with cytokine release syndrome. Thus, although the cytokine profile of the 4-1BB containing CAR iNKT cells was not always a typical Th1 profile, based on the cytotoxic effector function, little or no IL-6 production, and superior expansion levels, we think that 4-1BB signaling provides efficient and sufficient co-stimulation for optimal expansion and functions of iNKT cells.

In our study, we finally explored whether CAR iNKT cells can be propagated by α-GalCer loaded DCs, which are excellent stimulators of unmodified iNKT cells. We show that stimulation of CAR iNKT cells in a TCR-dependent fashion also enables expansion rates of 300–1000-fold for both BCMA- and CD38-CAR iNKT cells in 5 weeks.

Nonetheless, it needs to be emphasized that in our study we have not executed a full (clinical) scale CAR iNKT cell expansion, but we deduced these prospects based on expansion rates observed in small scale expansions starting with a very low number of iNKT cells. Thus, toward clinical applications, the expansion rates observed in this study need to be better determined in full-scale expansion experiments. Nonetheless, a recent interim report of the first CAR iNKT Phase I clinical trial, targeting GD2 in child neuroblastoma, demonstrated that CAR iNKT cells can be successfully expanded to administer 3 × 10⁶ CAR iNKT cells per square meter of body surface area, indicating the feasibility of such strategies [54]. Another remarkable observation which needs attention in future studies is the possible overstimulation of CD38-CAR iNKT cells through α-GalCer loaded DCs, because when stimulated this way, the percentages of CD38-CAR iNKT cells decreased in time, as shown in Figure 6, while this was not the case when they were stimulated in a CAR-dependent fashion, using CD1d low expressing UM9 cells. Since DCs express CD38 but not BCMA, overstimulation via combined CAR as well as TCR stimulation could constitute a plausible explanation for this observation using CD38-CAR iNKT cells, a phenomenon which was also demonstrated for CAR-transduced alloreactive T cells [12].

Finally, another advantage of stimulating CAR-iNKT cells via their invariant TCR may be the possibility of using α-GalCer loaded DCs to boost them in vivo by administration of α-GalCer loaded DCs into patients who have already been treated with CAR-iNKT cells. While this possibility needs to be tested in appropriate in vivo models, in this study we provide early ex vivo evidence, supporting this possibility by demonstrating that CD38-CAR iNKT cells which were continuously stimulated with tumor cells can still be expanded by simulation with α-GalCer loaded DCs.

In conclusion, our study demonstrates not only the feasibility but also the most optimal conditions for the generation of BCMA- and CD38-CAR transduced iNKT cells to maximally exploit the therapeutic properties of BCMA- and CD38-CARs iNKT cells in the battle against MM. iNKT cells can be safely used across HLA barriers and can be exponentially expanded ex vivo without losing their functional properties. This strategy holds great potential to apply to a broad patient population, hereby tackling an important drawback of CAR-based adoptive immunotherapy strategies.

## 4. Materials and Methods

### 4.1. CAR Constructs

CD38B1-CAR (in short CD38-CAR) constructs containing different costimulatory domains were previously described in detail [42,43]. The BCMA-CAR contained the 4-1BB costimulatory domain and was constructed using the published murine scFv sequence derived from BCMA02 CAR (product name bb2121, WO 2016/094304 A2) [55]. In the SFG retroviral construct, the scFv was followed by a CD8a transmembrane domain and the 4-1BB and CD3ζ signaling domains or a CD28 transmembrane domain and intracellular sequence as described in Zhao et al. [45]. The CAR sequences were linked by a P2A sequence [56] to a truncated low-affinity nerve growth factor (LNGFR), dsRed, or 4-1BBL sequence, as depicted in Figure 1A. The 4-1BBL sequence, which was obtained form EBV-LCL cell line 10850, was integrated in CD38B1-CAR vectors as described previously [43].

Phoenix-Ampho packaging cells were transfected with 10 μg CAR constructs + 5 μg gag-pol (pHIT60), and 5 μg envelope (pCOLT-GALV) vectors (Roche, Basel, Switserland) using calcium phosphate precipitation. Next, 16 h post-transfection medium was replaced with complete Dulbecco’s modified Eagle medium (DMEM) + 10% fetal bovine serum (FBS, and two and three days after transfection, cell free supernatants containing retroviral particles were collected and directly used for transduction of iNKT cells.

### 4.2. iNKT Cell Isolation, Expansion, and Transduction

Healthy donor blood peripheral mononuclear blood cells (PBMCs) were isolated from buffy coats (Sanquin, Amsterdam, The Netherlands) through Ficoll-Paque (GE Healthcare Life Sciences, Hoevelaken, The Netherlands) density centrifugation. iNKT cells were purified using a Vα24 specific iNKT isolation kit (Miltenyi Biotec, Leiden, The Netherlands). Isolated iNKT cells were cultured in RPMI-1640 (Life Technologies, Bleiswijk, The Netherlands) supplemented with 10% FCS (Invitrogen, Bleiswijk, The Netherlands), penicillin (100 U/mL) and streptomycin (100 µg/mL), and 5.5 mM beta-mercaptoethanol (Gibco, Bleiswijk, The Netherlands). After isolation, the iNKTs were stimulated with the negative PBMC fraction irradiated (25 Gray) in a 1:1 ratio, supplemented with 100 ng/mL α-galactosylceramide (α-GalCer) (KRN7000 Funakoshi, Tokyo, Japan). The culture was supplemented with rhIL-2 (50 U/mL) (R&D, Minneapolis, MN, USA), rhIL-7 (10 ng/mL), and rhIL-15 (10 ng/mL) (PeproTech, London, UK) twice a week until retroviral transductions, which were performed in a RetroNectin-coated (15 µg/mL) (Takara, Saint-Germain-en-Laye, France) non-tissue treated 24-well plate (Corning, Amsterdam, The Netherlands) at a density of 5–10 × 10^5^ cells/mL in the presence of 1.0 mL virus per well followed by spinoculation (3000 rpm, 1 h at room temperature) in the presence of 4 µg/mL Polybrene (Sigma, Zwijndrecht, The Netherlands). A second transduction was conducted after 16 h, replacing 2/3 of the cell supernatant with freshly obtained virus. Then, 6–8 h after the second hit, half of the cell supernatant was replaced by fresh complete culture medium (RPMI-1640) (Life Technologies, Bleiswijk, The Netherlands) with the cytokine mix as previously described. Transduction efficiency was determined after 72 h as described in the flow cytometry section. iNKT cells were cultured in 48 or 24 well plates and were split 1:2 once a cell density of 1.5–2 × 10^6^ cells per/mL was reached. Every 7–10 days, irradiated (25 Gray) α-GalCer pulsed dendritic cells (DC) were added at E/T ratio 1:1 and the cultures were supplemented with the cytokine mix described earlier.

### 4.3. Generation of moDCs

CD14^+^ cells were isolated from PBMCs using CD14 MicroBeads (Miltenyi Biotec, Leiden, The Netherlands) and were cultured in RPMI-1640 (Life Technologies, Bleiswijk, The Netherlands) supplemented with 10% FCS (Invitrogen, Bleiswijk, The Netherlands), penicillin (100 U/mL), and streptomycin (100 µg/mL) supplemented with 580 U/mL IL-4 and 1000 U/mL granulocyte/macrophage colony-stimulating factor (GM-CSF). After 5 days, 100 ng/mL α-GalCer and 1000 ng/mL Lipopolysacharide (LPS) were added. At Day 7, the mature, α-GalCer loaded moDCs with high expression of CD80, CD83, CD86, and HLA-DR were harvested and used to stimulate iNKT cells [57].

### 4.4. Primary Bone Marrow MNCs from MM Patients

Bone marrow mononuclear cells (BM-MNCs) from MM patients containing ~10–40% malignant plasma cells (determined as CD38^+^/CD56^+/−^/CD138^+^ cells by flow cytometry) were isolated from bone marrow aspirates through Ficoll-Paque (GE Healthcare Life Sciences, Hoevelaken, The Netherlands) density centrifugation. Isolated cells were directly used in cytotoxicity assays or cryopreserved in liquid nitrogen until use. All primary samples were obtained after informed consent and according to the code of conduct for medical research developed by The Council of the Federation of Medical Scientific Societies (FEDERA https://www.federa.org/codes-conduct).

### 4.5. Cell Lines

Luciferase-transduced or unmodified human MM cell lines UM9, MM1.s, and MM1.s-CD1d were previously described [41,58,59]. The MM cell lines were cultured in RPMI-1640 ((Life Technologies, Bleiswijk, The Netherlands)) with 10% HyClone FCS (Fisher Scientific, Amsterdam, The Netherlands) and antibiotics (penicillin; 100 U/mL, streptomycin; 100 µg/mL). Phoenix Ampho cells were cultured in Dulbecco’s modified Eagle medium (DMEM) + GlutaMAX ((Life Technologies, Bleiswijk, The Netherlands)) with 10% FCS (Invitrogen, Bleiswijk, The Netherlands) and penicillin (10,000 U/mL) and streptomycin (10,000 µg/mL).

### 4.6. Flow Cytometry

Flow cytometry was performed on an LSR Fortessa instrument (BD). Viable cells were determined with live/dead cell marker (LIVE/DEAD^®^ Fixable Near-IR; Life Technologies, Life Technologies, Bleiswijk, The Netherlands). Transduction efficiency via associated CAR expression was measured with an APC-conjugated antibody towards low-affinity nerve growth factor (LNGFR CD271), 4-1BBL (CD137L) (Biolegend, London, UK), or with F(ab’)2-Goat anti-Mouse (eBioscience, Bleiswijk, The Netherlands) in case of the murine-based BCMA sequence. CAR-28z-dsRed was measured in the PE-CF594 channel to detect dsRed. Additional antibodies used for iNKT phenotyping were: CD3, CD4, and CD8 (BD Bioscience, San Jose, CA, USA), Vα24, Vβ11 (Beckman Coulter, Woerden, The Netherlands), PD-1, CD57, CD39, Tim3, LAG3, CD62L (Biolegend, London, UK). In cytotoxicity assays antibodies against CD1d, CD3, CD19, CD38, CD56, and CD138 (BD Bioscience, San Jose, CA, USA) were used. Flow cytometry data were analyzed using FCS Express (Pasadena, CA, USA) V6 software.

### 4.7. Flow Cytometry-Based Cytotoxicity Assay

Serial dilutions of mock- or CAR-transduced iNKT cells were incubated with target cells for 16–24 h. To distinguish the two cell types, target cells or effector cells were pre-stained with 0.5 µM Violet tracer (Thermo Fisher, Bleiswijk, The Netherlands). After the incubation period, Flow-Count™ Fluorospheres (Beckman Coulter, Woerden, The Netherlands) were added and the cells were harvested and stained for different CD markers (see Section 4.6) to identify different cell subsets. Viable cells were then quantitatively analyzed through Flow-Count-equalized measurements. Percentage cell lysis was calculated as follows and only if the analyzed target cell population contained > 500 viable cells in the untreated samples. % lysis cells = (1 − ((#viable target cells in treated wells/#of beads)/(#viable target cells in untreated wells/#of beads))) × 100%.

### 4.8. Cytokine Measurements

CAR iNKT cells were stimulated with relevant target cells for 16 h. The cytokines (IL-2, IL-4, IL-6, IL-10, IL17A, TNF, and IFN-γ) released in the culture supernatants were quantified using a Cytokine Bead Array (CBA) Human Th1/Th2/Th17 cytokine kit (BD, San Jose, CA, USA) according to the manufacturers’ protocol.

### 4.9. Statistical Analysis

Statistical analyses were performed using GraphPad Prism software version 7.0 (GraphPad, San Diego, CA, USA). For normal distributions, parametric student’s *t*-tests were used. In analyses where multiple groups were compared, either a parametric ANOVA with Bonferroni post hoc test or nonparametric Kruskal–Wallis test were used with subsequent multiple comparison. A *p* value < 0.05 was considered significant.

## Figures and Tables

**Figure 1 ijms-22-01096-f001:**
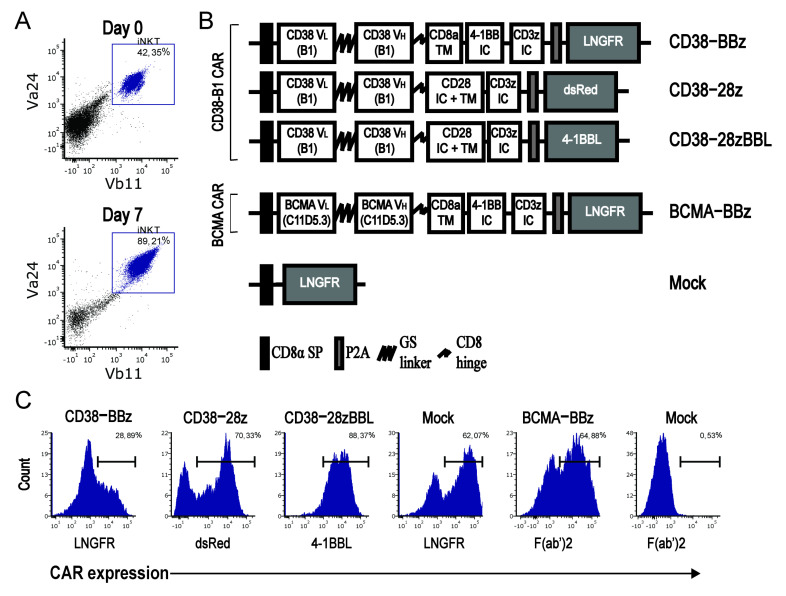
Invariant natural killer T (iNKT) cell isolation and CAR expression. (**A**) Representative dot plots depicting the gating strategy of iNKT cells by flow cytometry after purification with beads at Day 0 and at time of transduction on Day 7. (**B**) Schematic overview of different CD38- and BCMA-CAR (B cell maturation antigen-chimeric antigen receptor) constructs used; CAR expression is determined by expression of surrogate markers low-affinity nerve growth factor (LNGFR), dsRed, or 4-1BBL. (**C**) Flow cytometry histograms illustrating the surrogate marker expression of LNGFR and 4-1BBL as detected by APC-conjugated antibodies or by constitutive dsRed expression on the iNKT cells. The BCMA-CAR expression was determined by goat anti-mouse IgG polyclonal antibody targeting the murine sequence of the heavy and light chains of the CAR. Data are representative of independent transductions in iNKT cells of 3 donors for CD38-CARs and 6 donors for BCMA-CARs.

**Figure 2 ijms-22-01096-f002:**
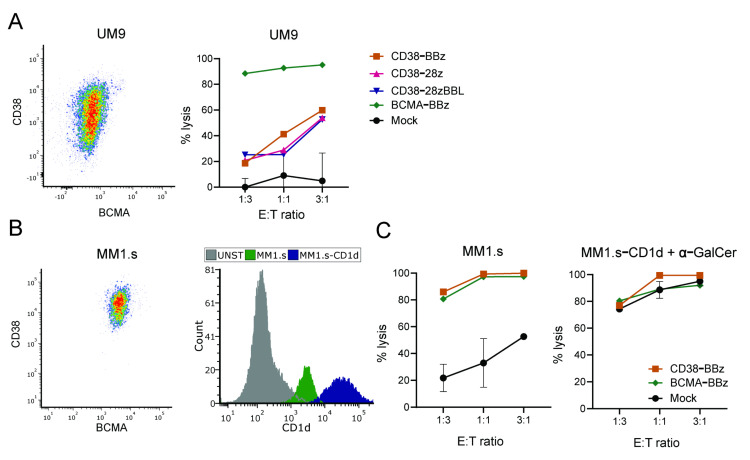
Cytotoxic capacity of iNKT-CARs against multiple myeloma (MM)-cell lines. MM cell lines were co-incubated with CAR iNKT cells at different E/T ratios as indicated for 16 h. (**A**) Flow cytometry density plot of UM9 depicting the expression of CD38 and BCMA and cytotoxicity with CD38-CARs with various co-stimulation domains and BCMA-CAR. (**B**) Flow cytometry density plot of MM1.s depicting the expression of CD38 and BCMA, histogram showing the expression of CD1d on MM1.s and MM1.s-CD1d cell line, and (**C**) cytotoxic activity of BBz-CAR iNKT cells on MM1.s cells after 16 h of co-incubation. Data is representative of 2 independent experiments. Error bars depict the SD.

**Figure 3 ijms-22-01096-f003:**
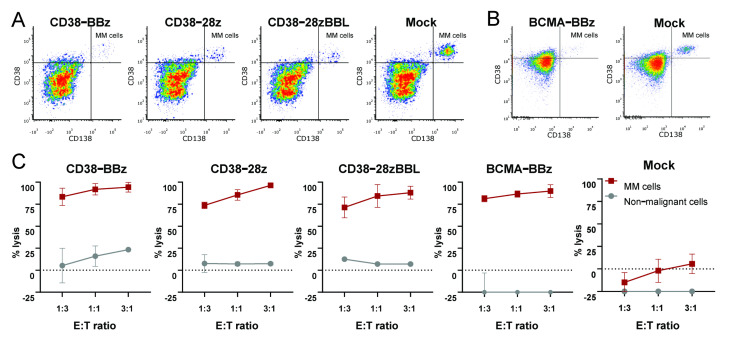
Effective on-tumor and minimal off-tumor cytotoxic activities of CD38 and BCMA CAR-iNKT cells in whole bone marrow samples. Representative flow cytometry density plots of two bone marrow samples depicting the specific eradication of MM cells identified as CD38^+^ CD138^+^ cells by CD38-CAR iNKT (**A**) and BCMA-BBz CAR iNKT (**B**) cells in 16 h assays. (**C**) Cytotoxic results of CD38- and BCMA-CAR iNKT cells in various E/T ratios. The error bars indicate the standard error of the mean (SEM )of independent assays with primary bone marrow mononuclear cell (BM-MNC) samples derived from different MM patients (CD38-BBz, CD38-28z, CD38-28zBBL *n* = 2, BCMA-BBz *n* = 6, Mock *n* = 8).

**Figure 4 ijms-22-01096-f004:**
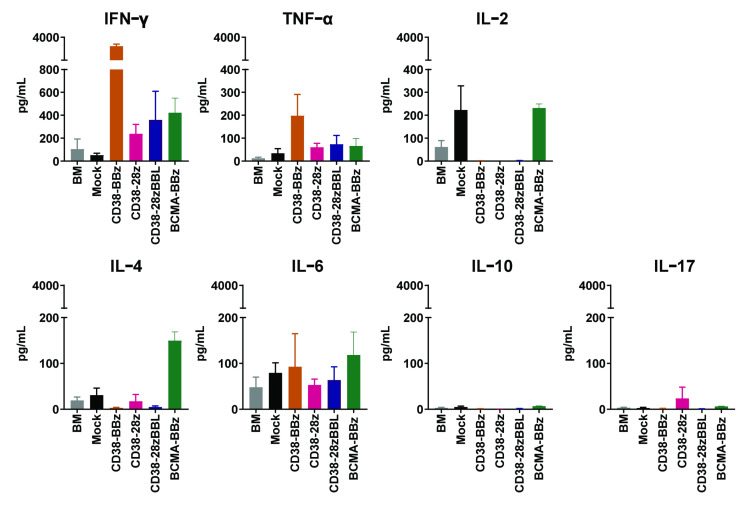
Cytokine production of CD38-CAR iNKT and BCMA-CAR iNKTs in response to primary BM-MNCs from MM patients. After co-incubation with BM-MNC samples, cell supernatants were harvested to measure cytokine secretion with a flow cytometry-based assay. Graph shows the secretion of IFN-γ, TNF, IL-2, IL-4, IL-6, IL-10, and IL17. Error bars indicate standard error of the mean (SEM) (*n* = 3 independent CAR iNKT batches, generated from three healthy donors).

**Figure 5 ijms-22-01096-f005:**
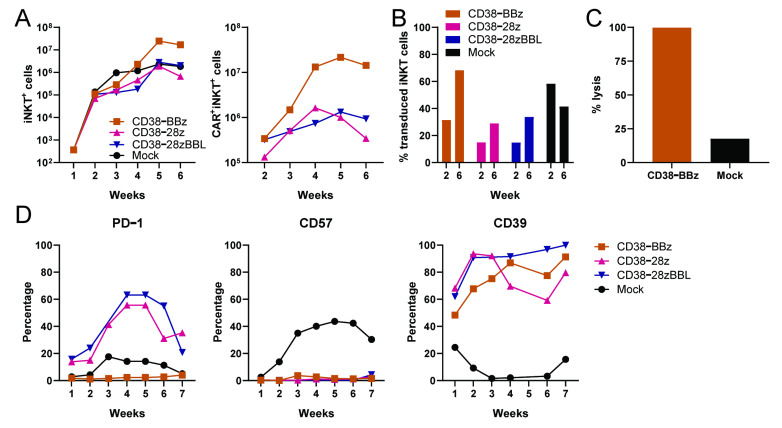
CD38-dependent expansion and activation of functional CD38-CAR iNKT cells. CD38-CAR iNKT cells with different costimulatory domains and mock-transduced cells were weekly re-stimulated with irradiated CD38^+^UM9 cell line at a CAR^+^iNKT:/target ratio of 3:1, starting 4 days after transduction. (**A**) Depicted are the calculated numbers of total iNKT cells (left) and of CAR ^+^ iNKT cells (right). (**B**) The percentage of CAR^+^iNKT cells in the culture at Week 2 and Week 6. (**C**) Cytotoxic activity of CD38-CAR vs mock-transduced iNKT cells towards a CD1d negative primary MM cell sample at the end of the cultures. (**D**) Graphs depict the expression of PD-1, CD57, and CD39 on iNKT cells. Similar results were observed in a second independent experiment.

**Figure 6 ijms-22-01096-f006:**
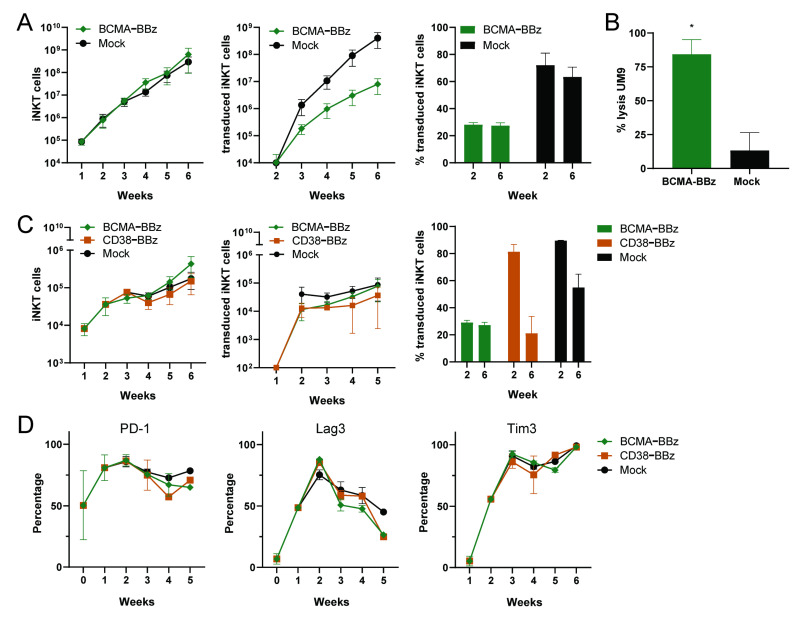
Expansion of functional CAR iNKT cells stimulated by α-GalCer loaded dendritic cells (DCs). BCMA-CAR iNKT cells and mock-transduced iNKT cells were weekly co-cultured with irradiated α-GalCer loaded DCs at an iNKT/DC ratio of 5:2, starting 4 days after transduction. (**A**) The calculated numbers of total iNKT cells (left), of transduced iNKT cells (middle), and the percentage of transduced iNKT cells at Week 2 and Week 6 of the culture (right). Error bars represent standard error of the mean (SEM) (*n* = 4 donors). (**B**) The cytotoxic activity of BCMA-BBz CAR and mock-transduced iNKT cells against UM9 cells at an E/T ratio of 3:1. Error bars represent SEM (*n* = 2) * *p* < 0.05 in a paired students *T* test. (**C**) Side by side comparison of CD38-BBz and BCMA-BBz iNKT cell expansion. Graphs show the calculated numbers of total iNKT cells (left), transduced iNKT cells (middle), and the percentage of transduced iNKT cells at Week 2 and 6 of cultures (right). Error bars represent SEM (*n* = 2 donors). (**D**) The expression of exhaustion markers PD-1, Lag3, and Tim3 during culture (*n* = 2).

**Figure 7 ijms-22-01096-f007:**
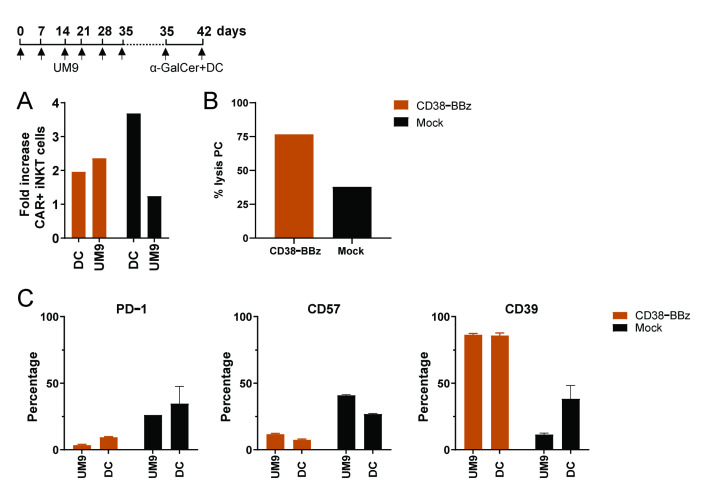
α-GalCer loaded DC induced expansion and activation of functional CD38-BBz CAR iNKT cells. Depicted is the culture scheme (upper panel **A**) and the proliferation at Day 42 (lower panel **A**) of CD38-BBz CAR iNKTs stimulated with CD38^+^UM9 cells until Day 35, followed by stimulation with α-GalCer loaded DCs. (**B**) Cytotoxic activity of CD38-CAR and mock-transduced iNKT cells towards primary MM cells at the end of the cultures. (**C**) The percentage of iNKT cells expressing activation/exhaustion markers PD-1, CD57, and CD39. Error bars depict standard error of the mean (SEM) (*n* = 2).

## Data Availability

Data is contained within the article or Appendix A.

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
