# Peer review of "Preclinical Evaluation of Invariant Natural Killer T Cells Modified with CD38 or BCMA Chimeric Antigen Receptors for Multiple Myeloma"

_ijms, 2021, doi:10.3390/ijms22031096_

Round 1
Reviewer 1 Report
Poels et al studied the potential of CD38 and BCMC-CAR iNKT cells for multiple myeloma in vitro. The authors characterized the CAR expression and efficiency of different iNKT cells and tested the cytotoxic capacity of these cells in MM1.s, UM9 cells and bone marrow samples from multiple myeloma patients. In addition, the authors identified the a-GalCer loaded DCs can facility to reserve CAR- or TCR-dependent cytotoxicity while expanding these iNKT cells. This is an interesting manuscript which might be applicable for preclinical use. The potential of these iNKT cells remains to be evaluated in an animal multiple myeloma model instead of focusing on in vitro or ex vivo test, although current datasets appear to be promising. It is highly recommended that the authors resubmit the manuscript for consideration after further in vivo studies are completed.
Author Response
Please see attachement.

Reviewer 2 Report
Poels and Drent et al describe in vitro evaluation of anti-tumor activity of invariant NKT cells with CARs against CD38 and BCMA. They report that the killing of MM cells by CAR-iNKT cells nicely correlates with the expression levels of CD38 and BCMA. They also provide conditions for in vitro expansion of these cells. It is an interesting study which is clear to follow.
The results are of some interest to researchers working with iNKT cells. I understand that the study design is solely in vitro, however, in vivo evaluation of these CAR loaded cells will be of greater interest and of broader clinical significance.
I only have a few minor suggestions:
-Lines 66-67 and 82-84 is confusing to the reader
-line 88 needs re-formatting
-words like ex vivo, in vivo, in vitro are not italicized uniformly; please make it uniform
-LNGFR abbreviation requires expansion somewhere in the manuscript
-In figure 3A, please provide at least one representative plot for CD38-28z, CD38-28zBBL, BCMA-BBz and Mock panels. Figure 3C is mentioned (line 161) but missing in the manuscript
-line 175 mentions “one of the primary MM samples” while figure 4 legend says n=3 different patient samples. Please clarify.
-Please replace the percentage symbol in y-axis of Figs 6D and 7C with the word
-Did the authors look at stimulating CAR-iNKTs themselves with a-GalCer?
This is an important experiment; possibly even restimulation with a-GalCer will be interesting.
Reviewer 3 Report
In this article Renée Poels et al. show a preclinical evaluation of Invariant Natural Killer T cells modified with CD38 or BCMA Chimeric Antigen Receptors for Multiple Myeloma. The authors have a considerable previous experience with Chimeric Antigen Receptor engineered T Cells.
In this research iNKT cells are transduced with a high affinity BCMA-CAR and with an affinity-optimized CD38-CAR, which selectively recognizes the high levels of CD38 antigen expressed on Multiple Myeloma (MM) cells due to its attenuated CD38 affinity. Both BCMA- and CD38-CAR transduced iNKT cells effectively redirected the killer functions of iNKT cells toward MM cells. The authors also test different coestimulatoy domains showing that 4-1BB provided the most suitable costimulatory signals. When both transduced cells are stimulated with the generic iNKT ligand, Alpha-galactosyl ceramide, loaded on monocyte-derived dendritic cells, they proliferate and maintain their cytotoxic capacity, independent of the CAR chain. These studies open the possibility of a new generic therapy against multiple mieloma.
Author Response
We thank the reviewer that he was satisfied with the manuscript and had no comments.
Round 2
Reviewer 1 Report
Recommended for publication in this journal.